# Insights into the Effects of Mesenchymal Stem Cell-Derived Secretome in Parkinson’s Disease

**DOI:** 10.3390/ijms21155241

**Published:** 2020-07-23

**Authors:** Michele d’Angelo, Annamaria Cimini, Vanessa Castelli

**Affiliations:** Department of Life, Health and Environmental Sciences, University of L’Aquila, 67100 L’Aquila, Italy; michele.dangelo@univaq.it (M.A.); annamaria.cimini@univaq.it (A.C.)

**Keywords:** secretome, Parkinson’s disease, stem cells, mesenchymal stem cells, exosomes, extravesicles, conditioned medium

## Abstract

Mesenchymal stem cell (MSC)-derived secretome demonstrated therapeutic effects like those reported after MSCs transplantation. MSC-derived secretome may avoid various side effects of MSC-based therapy, comprising undesirable differentiation of engrafted MSCs and potential activation of the allogeneic immune response. MSC-derived secretome comprises soluble factors and encapsulated extravesicles (EVs). MSC-derived EVs comprise microvesicles, apoptotic bodies, and exosomes. In this review, we focus on the recent insights into the effects of MSC-derived secretome in Parkinson’s disease (PD). In particular, MSC-derived secretome and exosomal components counteracted neuroinflammation and enhanced antioxidant capacity and neurotrophic factors expression. In light of the insights reported in this review, MSC-derived secretome or their released exosomes may be used as a potential therapeutic approach or as adjuvant therapy to counteract the disease progression and improve PD symptoms. Also, MSC-derived secretome may be used as a vehicle in cell transplantation approaches to enhance the viability and survival of engrafted cells. Furthermore, since exosomes can cross the blood–brain barrier, they may be used as biomarkers of neural dysfunction. Further studies are necessary to fully characterize the bioactive molecules present in the secretome and to create a new, effective, cell-free therapeutic approach towards a robust clinical outcome for PD patients.

## 1. Introduction

Several disorders, including neurodegenerative diseases, are in the focus of stem cell-based research. Mesenchymal stem cells (MSCs) are the most encouraging source for stem cell-based treatment thanks to their immuno-modulatory characteristics, pro-angiogenic features, and multi-lineage differentiation capability [1,2]. MSCs can be easily isolated from various sources, including adipose tissue, umbilical cord Wharton ’s Jelly, bone marrow, and dental pulp [1], which has encouraged numerous researchers to investigate their usage in cell transplantation approaches for Parkinson’s disease (PD).

Recently, the positive effects of stem cell transplantation have been ascribed to their secretome, composed of released bioactive factors, which offer a regenerative microenvironment for damaged tissues, triggering a self-regulated regenerative response and limiting the area of the lesion [3,4]. In particular, recent studies have aimed at the therapeutic potential of the secretome of MSCs. MSC-derived secretome comprises soluble factors and encapsulated extravesicles (EVs) [5,6]. Increasing evidence indicates that EVs have a strong impact on physiological processes and are particularly critical in cell-to-cell communication [7]. Various categories of vesicles have been defined, which show different properties and biogenesis. MSC-EVs contain membranes and cytoplasmic constituents of the original cells. MSC-EVs’ membranes are rich in sphingomyelin, cholesterol, and ceramide. They are positive for surface markers of MSCs (CD90, CD13, CD44, CD29, CD105, and CD73) but negative for the hematopoietic system-related markers (CD45 and CD34). Additionally, MSC-EVs express CD63 and CD81, typical markers of EVs [8,9,10].

MSC-derived EVs are composed of a lipid bilayer supplemented in proteins (integrins, tetraspanins, ligands for cell surface receptors) supporting trafficking, adhesion, and endocrine effects of EVs [11]. Numerous MSC-derived bioactive molecules are enveloped by the bilayer membrane, including enzymes, genetic materials (DNA, RNA, microRNAs), signal transduction proteins, immunomodulatory signaling, and growth factors [4,12].

MSC-derived EVs comprise microvesicles, apoptotic bodies, and exosomes [13]. Apoptotic bodies are the largest EVs (diameter >1000 nm), which degenerate from the MSCs during apoptosis. MSC-derived microvesicles are nano-sized (diameter 100–1000 nm) EVs that expand by budding from the plasma membrane [14]. Exosomes are the smallest EVs (diameter 30–200 nm) that are derived from the inner budding of endosome membranes, named multivesicular bodies. The merging of multivesicular bodies with the plasma membrane leads to exosomes being released into the extracellular environment, where they perform their biological activities by regulating different cell signaling pathways in target cells [7].

MSC-derived conditioned medium (MSC-CM) includes the entire set of MSC-derived soluble molecules and vesicular components [15,16], including exosomes.

Numerous biological effects were reported in experimental studies upon MSC-CM and MSC-EVs administration [4,17]. Notably, MSC-derived secretome avoids several side effects of MSC-based therapy, including undesirable differentiation of engrafted cells and potential activation of the allogeneic immune response [2,4]. Other issues in the use of stem cell transplant are low survival and engraftment, tumor formation, and the long wait time for cell preparation and proliferation. In contrast, MSC-derived secretome can be quickly manufactured from commercial cell lines, preventing invasive cell collection processes [18,19,20]. CM-secretome can be produced, freeze-dried, packaged, and transported more easily [16]. Furthermore, the rejection problems of the recipient, that can occur using stem cells per se, are avoided thanks to the lack of cells [21]. Moreover, MSC-derived secretome offers a useful source of bioactive factors since its content may be assessed through the analogous method of conventional pharmaceutical agents [22].

In light of these biological and manufactural advantages compared to MSC-based treatment, the administration of MSC-derived secretome has been deemed as a novel, cell-free beneficial approach for treating numerous disorders, including Parkinson’s disease [4]. MSC-derived secretome has been partially characterized by protein array; however, it is still unknown which sets of factors or molecules are related to the positive regenerative effects.

Parkinson’s disease (PD) is one of the most frequent neurodegenerative disorders. The hallmark of this disease is the loss of dopaminergic neurons in the substantia nigra with consequent motor and non-motor disorders due to dopamine loss and, thus, nigrostriatal pathway degeneration [23,24]. Further, neuroinflammation and oxidative stress are involved [25,26]. The only available treatments can relieve symptoms, but, to date, there is no cure. The investigations reported so far on MSC-derived secretome treatments in PD experimental models indicated MSC-derived secretome is a promising and encouraging approach for this disorder (summary in Figure 1).

### 1.1. Cellular and Molecular Mechanisms of PD

The underlying mechanisms of PD are still unclear, however, both environmental and genetic factors are involved in the pathogenesis [23,27]. Initially, the PD-linked mutations related to α-synuclein (A53T) [28]; subsequently, various gene mutations were identified, the most common were PINK1, DJ-1, LRRK2, and Parkin. The main cause of PD is the dopaminergic neuronal loss in the substantia nigra, which is the primary cause of motor symptoms. Dopamine metabolism, neuroinflammation, mitochondrial dysfunction, oxidative stress, and protein degradation damage are implicated in the death of dopaminergic neurons [23,29,30]. In addition, the immune system is implicated, indeed, in post-mortem brain and cerebrospinal fluid of PD patients, pro-inflammatory cytokines, including IFN-γ, TNF-α, IL-6, and IL-1β, are found to be upregulated [31].

For instance, in an α-synuclein PD rat model, an increase in IL-1β, IFN-γ, and TNF-α concomitant with microglia activation was found [32]. Recently, a study reported that T-cells from PD patients recognize α-synuclein peptides as antigenic epitopes, suggesting the association of PD with specific major histocompatibility complex alleles [33].

Mitochondrial dysfunction and oxidative stress are due to the accumulation of oxidized dopamine, which lead also to α-synuclein deposits and lysosomal impairment in neurons in PD patients [34,35]. Post-mortem analyses in PD patients’ brains revealed enhanced levels of 4-hydroxyl-2-nonenal (HNE), a by-product of lipid peroxidation, 8-hydroxyguanosine, and 8-hydroxy-deoxyguanosine oxidation products in the substantia nigra of PD patients [36,37]. The link between oxidative stress and PD pathogenesis is also confirmed by PD animal models induced by neurotoxins, such as 6-hydroxydopamine or methyl-4-phenyl-1,2,3,6-tetrahydropyridine (MPTP), which generate reactive oxygen species (ROS) production and consequent dopaminergic neuronal death [38]. Although ROS formation represents a relevant factor in PD, the molecular and cellular mechanisms connecting dopaminergic neuronal death and oxidative stress are still unclear. The primary insults lead to aberrant ROS production, which attack all macromolecules, promoting oxidative injury and leading to physiological activities impairments. Consequently, defects in these macromolecules cause mitochondrial impairment and neuroinflammation, which in turn promote ROS and eventually neuronal death. These loops lead to oxidative stress-mediated progressive death of dopaminergic neurons, a leading role in the neurodegenerative progression [23].

Another key characteristic of PD are Lewy bodies, eosinophilic fibrillary intracellular deposits in neuronal bodies and appendages. The main component of Lewy bodies are proteins, polysaccharides, and fats, in particular α-synuclein, ubiquitin, parkin, neurofilaments, and synphilin. The underlying mechanism of the generation of these aggregates is still unclear [37] as is their role in neuronal death. Further, an aberrant proliferation of different glial cell types occurs, [38], thus leading to microglia activation involved in the neuroinflammation. The neuronal loss and Lewy bodies aggregation occur not only in the substantia nigra and tectum mesencephalic and basal nuclei, but also in the pedunculopontine nucleus, the locus coeruleus, parasympathetic and sympathetic postganglionic neurons, the dorsal motor nucleus of the vagal nerve, the cerebral cortices, the raphe nucleus, the olfactory bulbs, and the amygdala. Degeneration in these structures involve non-motor clinical symptoms development.

The ubiquitin-proteasome system, crucial for cell differentiation, DNA replication, and transcription upon endogenous and exogenous stimuli, is involved in the pathogenesis of sporadic forms of PD. In PD, three different gene mutations (*SNCA*, *PARK2*, and *UCHLI*) are directly correlated with impairments in this system, with decreased 26s proteasome complex activity found in the substantia nigra of PD patients, and consequent oxygenated proteins accumulation [37]. For instance, several intron SNPs (Single Nucleotide Polymorphism in Intronic Sequences) and dinucleotide polymorphism in the promoter region of SNCA have been reported, which influence the stability of mRNA and are associated with PD. [38,39].

Further, neurotrophic factors are implicated in PD pathogenesis. For instance, the brain growth factor (BDNF) and the glial growth factor (GDNF) are crucial for differentiation and normal functioning of dopaminergic neurons and influence dopamine metabolism in the substantia nigra. BDNF stimulates neuroregeneration and neuroprotection. In in vivo PD models, BDNF protected dopaminergic neurons from death and improved dopaminergic transmission and performance in motor tests [39]. It is relevant to clarify the underlying molecular and cellular mechanisms of PD to elaborate new approaches to the early diagnosis and treatment of this disorder.

### 1.2. Searching for New Biomarkers and Therapeutic Approaches for PD

Human fluids are encouraging sources of molecular biomarkers, which can be categorized into molecules (e.g., elevated levels of 8-hydroxydeoxyguanosine, a byproduct of DNA oxidation, in PD patients’ urine), proteins (e.g., protein aggregates), and RNAs (e.g., noncoding microRNAs) [40]. The potential in using biomarkers isolated from body fluids (i.e., serum, cerebrospinal fluid, urine, blood, saliva, plasma) is the possibility of screening different molecules at once, while the cons are the low levels of molecules and the heterogeneity [41,42]. Exosomes have attracted a lot of interest because they are able to overcome this issue. Exosomes can be obtained from all bodily fluids, and they have a complex cargo of different RNAs (including microRNAs, ribosomal RNAs, and long noncoding RNAs), lipids, proteins, and DNA that in part depend on the tissue of origin and health conditions [7,43]. Catalytically active enzymes like PTEN (phosphatase and tensin homolog), and bioactive lipids such as prostaglandins, can be transferred by exosomes to target cells [25,26]. Exosomes carry different proteins, called “exosome markers”, the majority of which are associated to their biogenesis [44]. They also carry transmembrane proteins that can help in the immunoselection of exosomes with a precise cellular origin, thus increasing the sensitivity of exosomes as biomarkers. It has been reported that in neurodegenerative disorders exosomes carry misfolded proteins, such as *α*-synuclein in PD [45]. Among the different biomolecules related to exosomes, miRNAs have attracted the most attention as biomarkers.

The most relevant insight in therapeutic approaches for PD concerns the administration of the dopamine precursor l-DOPA (l-3,4-dihydroxyphenylalanine), which is able to ameliorate PD-related symptoms, increasing the level of the neurotransmitter but not replacing or counteracting dopaminergic neuron death [46]. Despite these limitations, the different side effects and the lack of improvement in nondopaminergic symptoms (such as psychiatric disorders or cognitive impairment), l-DOPA is the currently available treatment for PD patient. Indeed, to date, there is no cure [47]. Thus, among the innovative therapeutic approaches, the use of stem cells has received particular interest.

## 2. The Fate of MSC-Derived Secretome

The biodistribution of secretome and the exosomal component upon administration in vivo is limited.

Numerous approaches have been utilized for in vivo tracking to establish EVs’ biodistribution upon systemic administration in various animal models [48,49]. An interesting comparative study assessed the biodistribution of bone marrow (BM)-derived secretome (labelled with a near-infrared lipophilic dye) in mice after different routes of administration and at different dosages. This was the first study focused on EVs’ biodistribution in vivo and the results underlined the importance for therapeutic research using MSC-derived secretome [48].

Near-infrared (NIR) dyes are best for in vivo applications because of their high signal/noise ratio [50]. EVs with superparamagnetic iron oxide nanoparticles are exploited for sensitive magnetic resonance analysis for high detection and resolution of tissues [51]. Notably, in a traumatic brain injury rat model, intravenous injection DiI-labeled MSC-derived exosomes was able to reach brain, liver, lung, and spleen [52]. Exosomes were found to home to the lesioned site. Intranasal administration was identified as one of the best routes of administration of EVs and exosomes because it led to higher brain accumulation at the lesioned brain area [53]. Biodistribution of systemically administered EVs is a dynamic process: EVs are rapidly (within 30 min) distributed in the spleen, liver, and lungs and then EVs are excreted by renal and hepatic processing in 1 to 6 h [54].

Concerning exosomes, upon systemic administration, they generally localize to the intestine, liver, spleen, and lungs of mice, where the mononuclear phagocyte system is active [48,55]. Indeed, macrophage-depleted mice showed slower clearance of exosomes from circulation with respect to control animals, suggesting the leading role of macrophages in exosome biodistribution [56].

### 2.1. Positive Effects of MSCs in PD

The first study reporting the potential of MSCs in PD demonstrated that the transplant of Wharton Jelly-derived MSCs was able to improve motor behaviors in a hemiparkinsonian rat model, indicating that the secretion of trophic factors mediated the rescue of the degenerating dopaminergic neurons [57]. Rat bone marrow-derived MSCs (BM-MSCs) intravenous administration ameliorated functional impairment and protected tyrosine hydroxylase (TH)-positive fibers in the striatum and substantia nigra in a PD rat model (6-OHDA lesioned) [58]. The authors detected chemotactic cytokine SDF-1α in the BM-MSC-derived secretome. They revealed that this cytokine inhibited the apoptosis in PC12 cells exposed to 6-OHDA, with a resultant increase of dopamine release from these cells [58]. In addition, Cova and his research group reported that human MSCs transplantation in the striatum of 6-OHDA lesioned rats protected dopaminergic neurons and induced neurogenesis, suggesting that MSCs in situ may help lesioned neurons thanks to the local release of soluble factors, such as BDNF [59].

These reports provide the idea that bioactive molecules released by MSCs exert neuroprotective and antiapoptotic effects. Indeed, other researchers focused on the use of MSC-derived secretome (in the form of conditioned media or an exosomal component) as a cell-free transplantation method for PD.

### 2.2. Positive Effects of MSC-Derived Conditioned Medium in PD

While the cell-free trophic consequences of MSCs transplant have been reported in different in vivo models, research on the therapeutic impacts of secretome on neurodegenerative disorders is still scant.

Notably, different studies described MSC-CM positive effects in PD. In vitro studies reported the neuroprotective activity of the secretome derived from human bone marrow-MSCs and human tooth germ stem cells, in 6-OHDA-treated SH-SY5Y cells and murine differentiated neural stem cells [60]. Another research group [61] investigated the viability of dopaminergic cells from different sources upon rat bone marrow-MSC-derived secretome treatment, suggesting that prostaglandin E2 receptors represent the principal factor of neuroprotective events.

Other interesting research assessed the neuroprotective effect of adipose-derived mesenchymal stem cells (ASCs)-CM on neurotrophins gene expressions and TH+ cell density in 6-OHDA-lesioned rats. ASCs secrete numerous neurotrophic factors and cytokines in CM, which protect neurons by antioxidative and trophic effects. ASC-CM protected dopaminergic neurons by preserving TH+ neurons and by increasing BDNF and neurotrophin-3 expression [62]. The neurotrophic factor BDNF is crucial for neuronal survival in the substantia nigra [63]. The ability of MSC-derived secretome to reduce extracellular α-synuclein both in vitro and in vivo was also reported, mainly mediated by matrix metalloproteinase-2 [64].

An exciting study in a PD rat model compared human MSC bone marrow-derived (hBMSCs) transplantation with hBMSC-derived secretome. hBMSC-derived secretome was able to protect dopaminergic neurons when compared to only hBMSCs, and to ameliorate behavioral performances. Further, hBMSC-derived secretome had more impact on neuronal differentiation and survival in vitro. Finally, this research group analyzed the secretome through proteomic analysis and showed that hBMSCs release exosome-related factors, including those related to the ubiquitin-proteasome and histone systems. This work offered essential understandings on the potential use of hBMSC-derived secretome as a therapeutic tool for PD [65].

More recently, another research group compared the therapeutic effect of stem cells with its conditioned medium in a PD rat model induced by rotenone. In particular, they used bone marrow mesenchymal stem cells (BMSCs). Biochemical, histological, and immunohistochemical parameters were significantly ameliorated in both BMSCs and CM-treated groups as confirmed by anti-nestin, anti-glial fibrillary acidic protein, and anti-α synuclein analyses. Interestingly, the results upon CM were more evident, almost reestablishing the normal histological architecture of the substantia nigra [66].

Mesenchymal stem cells derived from human exfoliated deciduous teeth (SHED)-CM showed promising potential in regenerative medicine. Specifically, Chen and collaborators examined the therapeutic effect of SHED-derived CM in a rotenone-induced PD rat model. Notably, SHED-derived CM was able to improve motor performance in PD rats, reducing neuroinflammation (Iba1 and CD4 levels), increasing TH amounts in the striatum, and decreasing α-synuclein levels in both the substantia nigra and striatum [67].

The effects of menstrual blood-derived mesenchymal stem cells (MenSCs)-CM were evaluated in an in vitro model of PD: SH-SY5Y neuroblastoma cell line treated with 1-methyl-4-phenylpyridinium (MPP^+^). Interestingly, MenSCs-CM was efficient against MPP^+^ induced oxidative stress and inflammation [68].

Another relevant therapeutic utilization of MSC-derived secretome for PD is its combination with cell replacement approaches [69,70]. For instance, it has been shown that the pre-treatment of embryonic dopaminergic neurons with rat BM-MSC-derived secretome improved survival in a PD rat model (6-OHDA). Yao and collaborators reported that transplantation of secretome-treated neural stem cells into PD rats led to improved motor tests and cognitive tests, which was associated with increased cell survival and differentiation of dopaminergic neurons in ventral tegmentum [71].

### 2.3. Positive Effects of MSC-Derived Exosomes in PD

As we mentioned above, the secretome released by MSCs contains different bioactive molecules, including exosomes. MSCs can produce a higher amount of exosomes than can other kinds of cells [72]. Methods to isolate exosomes from the MSC-conditioned medium have been widely developed.

MSC-derived exosomes are hypoimmunogenic (due to the lack of MHC-II and low expression of MHC-I) nanocarriers that comprise various immunoregulatory components. Exosomes have several advantages, they are able to cross the blood–brain barrier and blood capillaries, and are small enough to avoid being cleared by the reticuloendothelial system [73]. The mechanisms of cellular recognition and internalization are still unclear. Antigen recognition, adhesion, and free-floating are described as cellular recognition mechanism, while fusion, phagocytosis, micropinocytosis, and raft- and receptor-mediated endocytosis are indicated as exosomal internalization processes [74].

Exosomes have proven effective in direct MSCs transplantation, and their positive therapeutic effects have been shown in different disease models, in particular, they were beneficial for central nervous system pathologies. In a stroke animal model, MSC-derived exosomes, intravenously administered, stimulated angiogenesis and neurogenesis, neurite remodeling, and improved animal motor performances [75]. The same neuroprotective effect was shown in a traumatic brain injury model after MSC-derived exosomes administration; indeed, a reduction of neuroinflammation and better outcomes were reported [76]. Spinal cord injury rats, upon MSC-derived exosomes injection, showed reduced inflammation and increased neuronal regeneration [77,78]. In addition, in Alzheimer’s disease, the positive effects of MSC-derived exosomes with a particular impact on neuroplasticity were reported [79].

Concerning PD, SHED-derived exosomes were found to rescue dopaminergic neurons from 6-OHDA–induced apoptosis in vitro, providing a potential regenerative treatment for this disorder [80].

A research group demonstrated that human umbilical cord mesenchymal stem cell (hucMSC)-derived exosomes reduced apoptosis in SH-SY5Y cell culture. Further, in a PD rat model (6-OHDA lesioned), exosomes crossing the blood-brain barrier, reached the substantia nigra, decreased dopaminergic neurons loss and apoptosis, improved apomorphine-induced asymmetric rotation, and increased dopamine levels in the striatum [81].

Overall, the use of exosomes to treat PD is encouraging (Figure 2); however, the exact underlying mechanism is still unknown.

## 3. MSC-Secretome: miRNA Relevance and Theranostic Applications in PD

Secretome derived from MSCs, and in particular its miRNAs component, has also been indicated as a valuable tool for targeted therapies and diagnostics. miRNAs are a highly studied class of non-coding RNAs responsible for the regulation of different genes through RNA messenger degradation or inhibition of their translation [82]. Numerous miRNAs have been indicated as α-synuclein modulators. For example, altered binding between and fibroblast growth factor 20 (FGF20) mRNA and miR-433 induced increased FGF20 levels, which consequently led to elevated α-synuclein protein levels in the cell [83]. Further, high miR-16-1 levels block the translation of the HSP70 (heat shock protein 70) mRNA, involved in α-synuclein inhibition, thus leading to an accumulation of α-synuclein [84]. Moreover, targeting miR-7, miR-153, and miR-34b/c from binding on their α-synuclein induces elevated levels of α-synuclein [85,86].

In PD, a relation between miR-34b/c reduction and the resultant DJ-1 and PARKIN decline in several brain areas was found [87]. Notably, increased miR-494 and miR-4639-5p levels trigger a direct decrease of DJ-1 protein expression, making dopaminergic neurons more susceptible and predisposed to the PD phenotype [88,89]. Interestingly, it has been recently proposed that MSC-derived secretome can ameliorate different biomarkers of PD pathophysiology, thus suggesting MSC-derived secretome as a promising approach to identify and generate valuable PD biomarkers [90]. As mentioned above, MSCs secrete different biomolecules and factors, including exosomes carrying miRNA, which may represent potential biomarkers but also modulators of different pathways underlying various disorders, including PD. Theranostic applications in PD exploiting the potential of MSC-derived secretome, mainly concern the targeting of the injured brain areas and delivering miRNAs through the blood-brain barrier. Thanks to the nature of exosomes, their application in the theranostic field and in clinic have received a lot of interest. Still, different points need to be addressed, such as the “best” MSCs line and the development of valid isolation techniques and loading methods without altering the exosomal component and integrity. However, MSC-derived exosomes may represent a valuable solution. In fact, numerous experiments revealed that MSC-derived exosomes are able to transfer miRNAs to neuronal cells, for instance, exosomes enriched in miR-133b can stimulate neurite outgrowth [90,91], one of the miRNAs generally decreased in PD. Further, miR-21 and miR-143, leading players in immune response, and neuroinflammation were also observed in MSC-derived exosomes [92]. Interestingly, in MSC-derived exosomes, a miRNA cluster composed of miR-18a, miR-17, miR-20a, miR-19a/b, and miR-90a, involved in axonal growth, neurogenesis, neurite remodeling and in CNS (central nervous system) recovery, was detected [93,94].

Another theranostic application in PD exploits the human Periapical Cyst-MSCs (hPCyMSCs) differentiated in dopaminergic neurons; thus hPCyMSC-derived exosomes may be useful therapeutic carriers for PD. hPCy-MSCs exposed to a neural-inductive medium led to functional dopaminergic neurons; the exosomes isolation from the CM of these MSCs is presently standardized [95]. The analysis of circulating exosome-derived miRNA through microarrays and gene sequencing could be related to nanotechnologies: This is a significant point to improving the capability of new smart nanomaterials to capture the small-sized biomolecules, representing a theranostic approach with elevated sensitivity and extreme specificity [96].

On this basis, it is relevant to isolate and characterize the entire set of biomolecules released by MSCs and in particular hPCy-MSCs, and to dissect the cellular and molecular mechanisms regulated by miRNAs. These new understandings may allow for the development of new therapeutic approaches and offer novel evidence on functional biomarkers for early diagnosis and monitoring of neurodegenerative diseases, with particular attention to PD.

## 4. Conclusions and Future Perspectives

PD is a debilitating neurodegenerative disorder that affects millions of people worldwide; however, the molecular and cellular underlying mechanisms are still unclear. Although there are advances in the PD research field, the current therapeutic approaches improve PD patients’ quality of life, but they are not able to counteract PD progression and to stimulate dopaminergic neurons survival/differentiation. Thus, recently, MSC-derived secretome and its exosomal components have been suggested as promising therapeutic tools for numerous neurodegenerative disorders, including PD, due to their ability to promote dopaminergic neurons survival, stimulate neurogenesis, decrease neuroinflammation, promote functional recovery in in vivo models. 

To date, there is no cure for PD, and thus, recently, attention has been focused on cell-free approaches. MSCs have become widely used for cell-based therapy due to less scientific and ethical issues compared to the use of other kinds of cells. The ability of MSCs to release exosomes and various trophic factors makes the use of MSCs attractive for PD treatment. Thanks to their small size and/or soluble nature, these secreted molecules can cross the blood–brain barrier; moreover, exosomes are intrinsically less risky compared to live stem cell transplants. Exosomes cannot transform into harmful or malignant cells; they cannot replicate; they are less prone to activate an immunogenic response; and a virus cannot infect them. In light of the insights reported in this review, the use of MSC-derived secretome is encouraging in PD.

Further studies are needed to identify a personalized approach for the different neurodegenerative diseases and to create a new, useful, cell-free therapeutic approach towards a robust clinical outcome for PD patients. Another point that needs to be clarified is if the encouraging results are due to one or two factors or a combination of different molecules present in the secretome. To date, it is pretty clear that MSC-derived secretome exerts positive effects on neuronal cell survival, differentiation, and proliferation; however, future studies need to characterize all the bioactive molecules fully. Thus, MSC-derived secretome or their released exosomes may be used as a potential therapeutic approach or as adjuvant therapy for PD symptoms and to counteract the disease progression. Furthermore, their secretome may be used as a vehicle in cell transplantation approaches to improve the viability and survival of engrafted cells and also as a diagnostic approach. These different aspects of the knowledge about the secretome may permit the advancement of targeted secretome to fight different pathophysiological impairments in a multidisciplinary manner. In addition, since MSC-derived secretome is able to stimulate neurotrophic (i.e., BDNF, a biomarker of the majority of neurodegenerative disorders) and neuronal survival pathways and to counteract neuronal death, it could also be beneficial against other neurodegenerative conditions, including polyglutamine disorder, Alzheimer’s disease, and stroke.

## Figures and Tables

**Figure 1 ijms-21-05241-f001:**
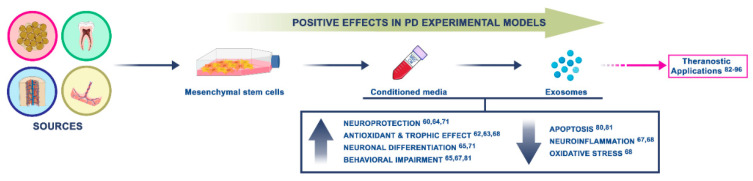
Summary of the encouraging insights on mesenchymal stem cell (MSC)-derived secretome treatment in Parkinson’s disease (PD) experimental models. In particular, we focused on the neuroprotective effects of conditioned media and exosome-derived MSCs.

**Figure 2 ijms-21-05241-f002:**
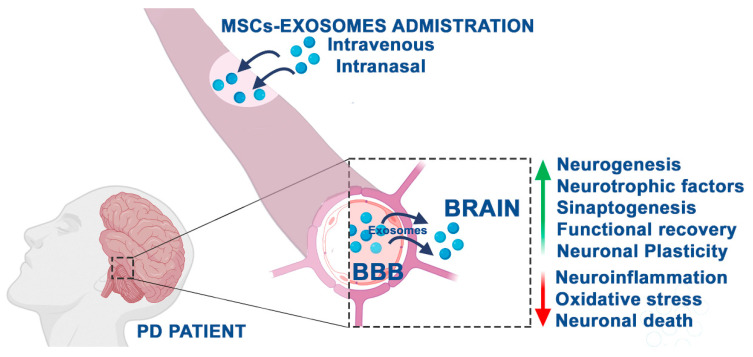
Therapeutic potential of MSC-derived exosomes in PD patients based on the results reported so far in experimental models. Through intravenous or intranasal administration, exosomes are able to cross the blood–brain barrier (BBB), thus exerting neuroprotective activities in neurodegenerative diseases, including PD.

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
