# Peer review of "Insights into the Effects of Mesenchymal Stem Cell-Derived Secretome in Parkinson’s Disease"

_ijms, 2020, doi:10.3390/ijms21155241_

Round 1

Reviewer 1 Report

This is a timely review by d’Angelo et al summarizing recent achievements in a therapeutic approach in PD based on MSCs derived secretome. It gives a good general background on PD and summarizes well literature that is related to this topic. The manuscript flows well overall. There are some minor issues to be addressed by authors:

Line 43: Authors should cite original papers rather than two review papers.

Line 65: The statement about the rejection is a bit controversial as it suggests that presence of cells can only trigger rejection process. I would suggest re-phrasing this sentence.

Line 157 It would be beneficial to readers if authors provide more details about the fate of MSC-derived secretome.

Line 303 Conclusions should be expanded to an application of MSCs secretome and exosomes to other neurodegenerative diseases including polyQ diseases as for example BDNF is a biomarker for majority of neurodegenerative diseases, not limited to PD.

Finally Figure2 should be redesigned, properly described in the figure legend and enlarged as in the current form is barely readable.    

Author Response

This is a timely review by d’Angelo et al summarizing recent achievements in a therapeutic approach in PD based on MSCs derived secretome. It gives a good general background on PD and summarizes well literature that is related to this topic.

The manuscript flows well overall. There are some minor issues to be addressed by authors:

We really appreciated the Reviewer’s suggestions and we tried to address all the points raised.

Line 43: Authors should cite original papers rather than two review papers.

Thank you. We now replaced with proper citations.

Line 65: The statement about the rejection is a bit controversial as it suggests that presence of cells can only trigger rejection process. I would suggest re-phrasing this sentence.

We totally agree with the Reviewer and we rephrased the sentence accordingly.

Line 157 It would be beneficial to readers if authors provide more details about the fate of MSC-derived secretome.

We thank the Reviewer. We now provided more information.

Line 303 Conclusions should be expanded to an application of MSCs secretome and exosomes to other neurodegenerative diseases including polyQ diseases as for example BDNF is a biomarker for majority of neurodegenerative diseases, not limited to PD.

We really thank the Reviewer and we now expanded the conclusions according to Reviewer’s suggestion.

Finally Figure2 should be redesigned, properly described in the figure legend and enlarged as in the current form is barely readable.  

We totally agree with the Reviewer and we now redesigned the figure.

Reviewer 2 Report

The review article by d’Angelo et al focuses on the effects of the secretome derived from mesenchymal stem cells in Parkinson’s disease. They give information on the origin of the exosomes, and their interest in the treatment of Parkinson’s disease. They recall that no cure is still available in this disease which is associated with the degeneration of the dopaminergic neurons of the substantia nigra. They focus then on the material contained in exosomes including miRNAs and numerous factors conferring neuronal protection, regeneration, anti-inflammatory responses. They finally point out that exosomes cross the blood brain barrier and are perhaps safer than stem cells transplantation. The authors wish additional studies to fully characterize the material contained in exosomes.

The review is interesting, and its organization is fair. I have some issues that have to be fixed.

Major points:

  1. The text has to be ameliorated. It is not only grammar or misspellings. Several sentences are too long, and the writing of some ideas should be ameliorated. I will give some examples below:

Line 94: “impairment in PD neurons and in patients”. It should be PD patients

Line 110-113: too long and unclear: “Further, in PD development of gliosis have been found [38] (GRAMMAR?), specifically in striatum and substantia nigra an aberrant proliferation of different glial cell types occur (GRAMMAR?), thus leading to microglia activation, involved in the neuroinflammation and in the increased expression of major histocompatibility complex class II genes.” It should be cut.

Line 117: “and the amygdala”

Line 130: “and improved dopaminergic neurotransmission and motor tests”; shouldn’t be “improved dopaminergic transmission and the performances in motor tests”? The same comment applies line 228.

Line 151: “which is”

Line 154: “L-DOPA is the only available treatment for PD patient”. This statement is wrong.

Line 167: “distribute” or distributed?

Line 172-175. This sentence can surely be ameliorated

Line 181: the abbreviation of BDNF appears at several occasions; meanwhile, the text has to be checked for the abbreviations.  Another interesting example to illustrate this point, line 99: “which generate ROS production and consequent dopaminergic  neuronal death [36]. Although reactive oxygen species (ROS) formation”.

Line 192: represent

Lines 196-198: “its”? Note the BDNF here.

Line 281-284: too long

These are only few examples. The text should be read and corrected by a native English speaker.

  1. The figure 2 is placed at the end of the article. Since it is a general approach, it could be placed as figure 1 in the beginning of the text. Thereafter, the subsequent information in the text could make reference to this figure 1.

Author Response

The review article by d’Angelo et al focuses on the effects of the secretome derived from mesenchymal stem cells in Parkinson’s disease. They give information on the origin of the exosomes, and their interest in the treatment of Parkinson’s disease. They recall that no cure is still available in this disease which is associated with the degeneration of the dopaminergic neurons of the substantia nigra. They focus then on the material contained in exosomes including miRNAs and numerous factors conferring neuronal protection, regeneration, anti-inflammatory responses. They finally point out that exosomes cross the blood brain barrier and are perhaps safer than stem cells transplantation. The authors wish additional studies to fully characterize the material contained in exosomes.

The review is interesting, and its organization is fair. I have some issues that have to be fixed.

We really appreciated the Reviewer’s suggestions and we tried to address all the points raised.

Major points:

  1. The text has to be ameliorated. It is not only grammar or misspellings. Several sentences are too long, and the writing of some ideas should be ameliorated. I will give some examples below:

Line 94: “impairment in PD neurons and in patients”. It should be PD patients

Line 110-113: too long and unclear: “Further, in PD development of gliosis have been found [38] (GRAMMAR?), specifically in striatum and substantia nigra an aberrant proliferation of different glial cell types occur (GRAMMAR?), thus leading to microglia activation, involved in the neuroinflammation and in the increased expression of major histocompatibility complex class II genes.” It should be cut.

Line 117: “and the amygdala”

Line 130: “and improved dopaminergic neurotransmission and motor tests”; shouldn’t be “improved dopaminergic transmission and the performances in motor tests”? The same comment applies line 228.

Line 151: “which is”

Line 154: “L-DOPA is the only available treatment for PD patient”. This statement is wrong.

Line 167: “distribute” or distributed?

Line 172-175. This sentence can surely be ameliorated

Line 181: the abbreviation of BDNF appears at several occasions; meanwhile, the text has to be checked for the abbreviations.  Another interesting example to illustrate this point, line 99: “which generate ROS production and consequent dopaminergic  neuronal death [36]. Although reactive oxygen species (ROS) formation”.

Line 192: represent

Lines 196-198: “its”? Note the BDNF here.

Line 281-284: too long

These are only few examples. The text should be read and corrected by a native English speaker.

We really appreciated the Reviewer’s comments that helped to improve our manuscript. We checked thoroughly the manuscript as suggested and made substantial changes.

  1. The figure 2 is placed at the end of the article. Since it is a general approach, it could be placed as figure 1 in the beginning of the text. Thereafter, the subsequent information in the text could make reference to this figure 1.

We thank the Reviewer for the suggestion and we totally agree. We now placed the Figure 2 in the beginning of the manuscript.

Round 2

Reviewer 2 Report

The review article by d’Angelo et al focuses on the effects of the secretome derived from mesenchymal stem cells in Parkinson’s disease. They give information on the origin of the exosomes, and their interest in the treatment of Parkinson’s disease. They recall that no cure is still available in this disease which is associated with the degeneration of the dopaminergic neurons of the substantia nigra. They focus then on the material contained in exosomes including miRNAs and numerous factors conferring neuronal protection, regeneration, anti-inflammatory responses. They finally point out that exosomes cross the blood brain barrier and are perhaps safer than stem cells transplantation. The authors wish additional studies to fully characterize the material contained in exosomes.

This is a revised version corresponding to the first submission and the authors made the appropriate changes. This version is better and acceptable for publication.